# Role of Decorin in the Lens and Ocular Diseases

**DOI:** 10.3390/cells12010074

**Published:** 2022-12-24

**Authors:** Eri Kubo, Shinsuke Shibata, Teppei Shibata, Hiroshi Sasaki, Dhirendra P. Singh

**Affiliations:** 1Department of Ophthalmology, Kanazawa Medical University, Kanazawa 920-0293, Ishikawa, Japan; 2Department of Ophthalmology, University of Nebraska Medical Center, Omaha, NE 68198, USA

**Keywords:** decorin, posterior capsule opacification, fibrotic diseses, TGF-β2, FGF-2

## Abstract

Decorin is an archetypal member of the small leucine-rich proteoglycan gene family and is involved in various biological functions and many signaling networks, interacting with extra-cellular matrix (ECM) components, growth factors, and receptor tyrosine kinases. Decorin also modulates the growth factors, cell proliferation, migration, and angiogenesis. It has been reported to be involved in many ischemic and fibrotic eye diseases, such as congenital stromal dystrophy of the cornea, anterior subcapsular fibrosis of the lens, proliferative vitreoretinopathy, et al. Furthermore, recent evidence supports its role in secondary posterior capsule opacification (PCO) after cataract surgery. The expression of decorin mRNA in lens epithelial cells in vitro was found to decrease upon transforming growth factor (TGF)-β-2 addition and increase upon fibroblast growth factor (FGF)-2 addition. Wound healing of the injured lens in mice transgenic for lens-specific human decorin was promoted by inhibiting myofibroblastic changes. Decorin may be associated with epithelial–mesenchymal transition and PCO development in the lens. Gene therapy and decorin administration have the potential to serve as excellent therapeutic approaches for modifying impaired wound healing, PCO, and other eye diseases related to fibrosis and angiogenesis. In this review, we present findings regarding the roles of decorin in the lens and ocular diseases.

## 1. Introduction

Decorin (DCN) belongs to a growing family of class I small leucine-rich proteoglycan proteins (SLRPs) that consist of a single glycosaminoglycan side chain (GAG), including a core protein containing leucine-rich repeats (LRR) with a molecular weight of approxi-mately 42 kDa [1]. Decorin is mainly synthesized by fibroblasts, stressed vascular endo-thelial cells, and smooth muscle cells. It is widely distributed in the stromal tissues and is expressed and secreted in various connective tissues, including the skin, bone, cartilage, ligaments, breast and vessel walls, and eyeballs, and in body fluids such as the plasma, aqueous humor, and vitreous humor [1,2,3,4,5,6,7,8,9,10,11,12,13,14]. Decorin is the main component of the extracellular matrix (ECM), together with collagen fibrils, and acts as a bridging molecule between type I and type VI collagen [15,16,17]. Decorin also regulates various cellular processes, including the ECM remodeling, fibrosis, cell proliferation, migration, differentiation, wound healing, autophagy, and angiogenesis. It also has essential inhibitory roles in inflammation, fibrotic disorders, and tumors [3,6,18,19,20,21,22,23,24,25,26]. Decorin interacts with and modulates multiple growth factors and signaling pathways, such as the transforming growth factor (TGF)-β pathway, fibroblast growth factor (FGF), platelet-derived growth factor (PDGF), and cellular communication network factor 2/connective tissue growth factor (CCN2/CTGF). Moreover, it directly blocks various receptor tyrosine kinase (RTK) family proteins, such as the epidermal growth factor receptor (EGFR), Erb4, and its other family members, insulin-like growth factor-I receptor (IGF-IR), vascular endothelial growth factor receptor 2 (VEGFR2) and the hepatocyte growth factor receptor 1 (HGFR1), and Toll-like receptors (TLRs) binding to multiple binds multiple cell surface receptors [4,5,26,27,28]. Decorin also acts as a reservoir for several different growth factors, sequestering signaling molecules in the ECM [29]. Thus, it is well known that decorin evokes potent tumor repression and wound-healing properties.

Here, we review the current knowledge of the role of decorin in the lens and ocular diseases. We hypothesize that decorin plays an important regulatory role in inhibiting ocular inflammatory and fibroblastic responses.

## 2. Structure and Function of Decorin

The structural and putative functional features of decorin are shown in Figure 1. The mature decorin molecule consists of a globular core protein and a covalently- linked GAG chain [30]. The primary translation product of human decorin is 359 amino acids long and can be divided into four functional domains (Figure 1) [30]. Domain I contains the initial 16-amino-acid long signal peptide directing the nascent protein to the rough endo plasmic reticulum, which is absent in the mature decorin protein [31]. Domain II contains 14 amino acid propeptide residues, regulating the attachment of GAGs composed of either dermatan or chondroitin sulfate [31]. GAG interacts with thrombospondins (Thbs), other GAGs, cytokines, and growth factors, such as TNF (tumor necrosis factor)-α and FGF1, 2, 7, and 8, regulating cell growth, wound healing, axon regeneration, and neural stem cell proliferation [32,33]. Decorin GAG may also inhibit blood coagulation by interacting with heparin cofactor II [34].

Domain III is the most important domain of decorin and has a major regulatory role by interacting with different signaling molecules [30]. Domain III consists of twelve tan-dem LRRs flanked by conserved cysteine (Cys)-rich domains and is important for its biological function. Decorin has a high-affinity binding site for collagen at LRRs 4–6 and a low-affinity site at the C-terminus for interacting with multiple collagen molecules simultaneously [31]. Decorin is also involved in the regulation of collagen fibrillogenesis. Mutant mice with a targeted disruption of the decorin gene (*Dcn*^−/−^) have fragile skin with abnormal collagen architecture in the dermis and reduced tensile strength, and showed a delay in epidermal and dermal wound healing [6,35,36]. Decorin has two binding sites for TGF-β isoforms at LRR4 and LRR5 with varying affinity in its core protein domain III, thus regulating TGF-β/ECM interactions (Figure 1) [37]. Decorin binds with high affinity to all isoforms of the TGF-β family and neutralizes their interaction with pro-fibrotic receptors [20]. LDL receptor-related protein 1 (LRP-1), an endocytic receptor for decorin involved in mediating TGF-β-dependent signaling, binds LRR5 and LRR6 [38,39,40]. CCN2/CTGF is an matricellular protein which that bridges the functional divide division between structural macromolecules and growth factors, which regulates regulating many cellular functions including cell adhesion, matrix production, structural remodeling, an-giogenesis, and cell proliferation, fibrosis, and differentiation [41]. CCN2/CTGF binds and induces growth factors (e.g., TGF-β and VEGF, thereby regulating their functions) and ECM proteins [41]. CCN2/CTGF also binds to LRR10-12, stimulating decorin and repressing the activity of important cytokines [42,43]. The LRR7 domain binds to EGFR and ErB4 and activates the mitogen-activated protein kinase (MAPK) pathway [44]. The LRR5 domain binds directly to VEGFR2 at a site that partially overlaps with the canonical binding site for VEGF-A [45]. Domain IV, the carboxyl terminus domain, contains binding sites for collagen fibrils and fibronectin (Fn) [30].

Although they are do not act as binding ligands for decorin, proteases play an im-portant role in the decorin interaction network. These proteases primarily include matrix metalloproteinases (MMP) 2, 3, and 7 [46,47,48]. MMP2, MMP3, and MMP7 induce the deg-radation of decorin, which is digested into fragments via interactions with its protein core [49]. This digestion of decorin allows the release of TGF-β1 [49]. MMP2 and MMP9 also activate TGF-β2 [50,51]. Thus, a single decorin molecule could simultaneously sequester multiple important mediators of cell growth and antagonize several signaling pathways crucial for fibrotic diseases. 

## 3. Interaction between Decorin and TGF-β

TGF-β is considered a potent and ubiquitous multifunctional cytokine involved in wound repair, proliferation, migration, apoptosis, differentiation, and fibrosis, including a pleiotropic effect in immune response [52,53]. The expressions of TGF-β is increased in almost all fibrotic diseases involving all organ systems, including ocular diseases [54,55,56,57,58,59,60]. Three isoforms of TGF-β exist: TGFβ1, TGFβ2, and TGFβ3. They are secreted by many types of cells and are involved in a wide range of actions, including the promotion of cell differentiation, promotion or inhibition of cell proliferation, induction of ECM production, immunosuppression, and increase in mitochondrial reactive oxygen species (ROS) production [59]. TGF-β is also considered a potent immunosuppressive cytokine and is considered a target for immunotherapy to suppresses tissue rejection and prolong corneal allograft survival after penetrating keratoplasty [53,61]. The intracellular signaling of the TGF-β family is regulated by Smads. TGF-β signaling through the canonical Smad pathway has been well-described for its regulation of multiple target genes, including fibrosis inducible genes [60]. Smad2 and Smad3 are the two major downstream regulators, while Smad7 serves as a negative feedback regulator of the TGF-β/Smad pathway [62]. The canonical TGF-β signaling pathway involves ligand-receptor complex formation, the activation of kinase domains within the cell membrane receptors and its subsequent phosphorylation cascade, and the activation of intracellular signaling downstream of Smad-2/3 [63], which are transcriptional regulators that accumulate in the nucleus. Thus, Smad-2/3 both transmit signals into the nucleus and regulate gene transcription by binding to Smad-binding elements present in the promoters of their target genes [63,64]. This phenomenon has been demonstrated in several cell types in vitro and in vivo [40,48,49]. Besides Smad pathways, other non-canonical pathways of TGF-β signaling, including the MAPK, phosphatidylinositol-3-kinase (PI3K), and Rho-like GTPase pathways, are also crucial for the effect of TGF-β on different pathways [65,66,67]. These non-canonical path-ways of TGF-β signaling indirectly regulate gene transcription in the nucleus as activators of Smad-dependent pathways [52]. Moreover, TGF-β upregulates epithelial–mesenchymal transition (EMT) related genes, such as fibronectin, α-smooth muscle actin (αSMA), and tropomyo-sin (Tpm) [7,68,69,70]. Decorin regulates the Smad pathway by interacting with LRP1, an endocytic receptor for decorin [44]. In addition, decorin can bind to TGF-β and inhibit the binding of TGF-β ligands to their receptors. The bound complex formation with decorin prevents the binding of TGF-β to its receptor, repressing its activity [30]. 

Many previous studies have confirmed the regulatory role of decorin for TGF-β activ-ity [71,72]. Our and other previous studies have demonstrated that TGF-β regulates gene transcription, and some genes, including the decorin gene, contain putative TGF-β-negative cis-acting elements (TIE) [73,74,75,76,77,78]. The promoter of decorin is divided into two main regions: a proximal promoter at 188 bp and a distal promoter at ~800 bp (Figure 2). The proximal promoter contains two functional TATA and CAAT boxes [79]. This region also contains two TNF-α-responsive elements (TNFRE) that mediate the TNF-α-induced transcriptional repression of the decorin gene [80]. The proximal promoter region also contains an activator protein 1 (AP1) binding site, a dual regulator of decorin gene expression repression by TNF-α and induction by interleukin-1 (NF/IL-1) [81]. The distal promoter of decorin has many homologous cis-acting factors, including AP1, AP5, and nuclear factor κ-light-chain-enhancer of activated B cells (NF-κB), several direct repeat factors, and a TIE [79]. It has been reported that the TIE at position −685 of the promoter region of the decorin gene exon Ib is a transcriptional silencer [76,77,78]. This TIE has been found in various protease gene sequences that are downregulated by TGF-β and could inhibit decorin gene transcription [78]. Supporting this regulatory mechanism, we reported that decorin expression was downregulated in lens epithelial cells (LECs) and rabbit articular chondrocytes after the addition of TGF-β2 [7,82]. On the other hand, the expression of decorin was reported to be upregulated in mesangial cells and rat liver cells, and after treatment with TGF-β [83]. These results show that the molecular mechanisms for decorin induction may differ among cell types and that many aspects of the underlying regulatory mechanisms are still currently unknown.

The ocular tissues in the eyes are susceptible to fibrotic diseases, such as primary an-terior subcapsular fibrosis (ASF) in cataracts, secondary posterior capsule opacification (PCO) after cataract surgery, scarring in the cornea and conjunctiva , primary open-angle glaucoma (POAG), age-related macular degeneration (AMD), proliferative diabetic reti-nopathy (PDR), and proliferative vitreoretinopathy (PVR) [2,57,84,85,86,87,88,89]. TGF-β signaling is involved in the pathogenesis of fibrotic eye diseases, impairing the quality of vision and intra- and extra-ocular homeostasis [2,57,84,85,86,87,88,89]. Physiologically, TGF-β is mainly pro-duced in the ciliary epithelium, LECs, and retinal pigment epithelial cells (RPEs) as a la-tent, inactive form of mature TGF-β [90,91,92,93]. TGF-β2 was also detected in human tears [94]. TGF-β2 levels change in the aqueous humor during the clinical course of various ophthalmic diseases. For example, changes in TGF-β2 levels in the eyes of patients with PVR could cause post-retinal detachment and retinal fibrosis. In addition, TGF-β2 concentrations in the vitreous fluid increase as retinal fibrosis progresses [95]. Levels of total and active TGF-β2 are higher in cataract patients with diabetic retinopathy, neovascular glaucoma, and POAG than in cataract patients without these ocular diseases [96,97,98,99]. Altered decorin expression is also associated with several fibrotic eye diseases, including congenital stromal dystrophy of the cornea [100,101], corneal wound healing [102], PCO [2,7], glaucoma [10,88], and PVR [103]. Thus, interactions between decorin and TGF-β may be involved in the pathogenesis of ocular diseases that involve fibrotic changes.

## 4. Distribution of Decorin in the Eye

Decorin is mainly localized to the connective tissue and extracellular matrix and primarily involves collagen fiber formation. In the eyes, decorin is expressed in all ocular tissues such as the corneal stroma, conjunctival stroma, trabecular meshwork, lens, and in all layers of the retina, optic nerve, and sclera. It is secreted in the tears, aqueous humor, and vitreous humor [1,2,7,9,13,14,88,103,104]. Decorin is typically synthesized and se-creted by fibroblasts, maintaining the balance of ECM homeostasis [105]. However, epi-thelial and endothelial cells can also synthesize decorin [105]. Recent work has shown that decorin expression is conisderably upregulated in LECs during PCO and pseudoexfoliation glaucoma (PXG), in the aqueous humor of subjects in PDR [2,7,9,106], and in the vitreous humor of patients with rhegmatogenous retinal detachment and PVR [103,107]. On the other hand, decorin protein levels were also markedly decreased in the aqueous and vitreous humor in patients with POAG [88]. We previously reported that the protein level of decorin was not altered in the aqueous humor and LECs with aging when we compared the grade and type of lens opacity using human anterior lens capsule samples obtained during cataract surgery [7]. The expression of decorin is modulated by various cytokines and can reduce inflammation and fibrosis. Therefore, the altered expression levels of decorin in eye diseases suggest that it may play a role in modulating the ECM and cellular and microvascular integrity.

## 5. The Role of Decorin in the Lens

No drug has been discovered to treat or prevent cataracts; thus, the only current treatment for cataracts is surgery. In modern cataract surgery, a lens capsular bag is produced after the aspiration and removal of the lens contents, consisting of a portion of the anterior and the entire posterior and intraocular lens (IOL) inserted into the lens capsule. However, after several years post-operation, patients may develop opacity in the lens posterior capsule. Due to PCO, visual function is impaired; this is referred to as “after-cataracts” [86]. Months to years after cataract surgery, fibroblast-like differentiation, EMT, and the formation of a regenerating lens (called Elschnig pearls) may develop, leading to clouding and shrinkage of the lens capsule [86,108]. EMT results in an increased cell motility and ECM accumulation in LECs and has also been reported to be implicated in cell invasion, wound healing, and fibrosis [57,109]. After cataract surgery, transdifferentiation of LECs into fibroblast-like cells is induced in residual LECs, expressing high levels of fibronectin, procollagen, Tpm, and αSMA, resembling the wound healing process [68,69,86,108]. 

Aberrant TGF-β signaling plays a central role in the pathobiology of human ASF in the lens [57,110,111,112] and PCO [68,69,108,113,114]. As with TGF-β, the biological effects of FGF2 on the development of PCO may be either detrimental or beneficial since FGF2 stim-ulates LEC proliferation and mitosis and collagen synthesis but reduces the expression of αSMA [115,116,117]. FGF2 stimulates lens fiber differentiation in a dose-dependent manner [118,119] and activates LEC mitosis, increasing collagen formation [120]. The concentra-tion of FGF2 increases in the LECs after cataract surgery [121]. Aberrant TGF-β2 signaling also promotes LEC migration, proliferation, stress fiber formation, and EMT, inducing the appearance of αSMA-positive myofibroblasts, which is modulated by FGF2 [68]. 

There are only a few reports on the role of decorin in the lens and its involvement in lens-related diseases. Under normal physiological conditions, the expression of decorin in LECs is low; however, the mRNA levels of decorin are increased in human LEC tissues (HLECs) with ASF obtained during cataract surgery [2]. Our recent study revealed that decorin may play a significant role in PCO and wound healing in the lens [7]. The expres-sion of many genes related to the ECM, including decorin, collagen, and fibronectin, were upregulated in LECs obtained from a mouse PCO model using a microarray-based ap-proach after extracapsular clear lens extraction (ECLE) [7]. Increased expression of decorin during EMT was confirmed in both rat and mouse PCO models via real-time quantitative PCR and Western blotting, suggesting that decorin is related to the cell proliferation, dif-ferentiation, fibrosis, and EMT of LECs observed in ASF and PCO [7]. Therefore, we ana-lyzed the interaction of decorin with TGF-β and FGF2 in the lens. In cultured mouse LECs (MLECs) and HLECs, the expression of decorin mRNA levels increased in response to FGF2 and decreased in response to TGFβ-2 [7]. Similar to our report, reduced expressions of decorin protein and mRNA after TGF-β treatment have been observed in many cell types, including fibroblasts [122,123]. It has been reported that the decorin gene contains TIE as a transcriptional silencer [76,77,78]. Thus, the addition of TGF-β may transcriptionally inhibit the expression of decorin in cultured LECs. 

On the other hand, the mechanisms underlying the transcriptional regulation of decorin by FGF have not been elucidated, and the effect of FGF2 on the expression of decorin differed among various cell types [124,125,126,127]. Previous studies reported that decorin is secreted from skin fibroblasts and binds to FGF2 and FGF-7 in the human wound fluids collected within 24 h of surgery, activating cellular proliferation through the activation of the MAPK pathway [128,129]. Thus, the interaction of FGF2 and decorin may induce cell proliferation in PCO tissues. hDCN overexpression in HLECs suppressed TGF-β-induced Tpm1 upregulation [7]. Tpms are known as proteins that stabilize F-actin filaments, regulating the dynamic and structural properties of these filaments by controlling their interactions with actin binding proteins [130,131]. In our previous study, we observed that *Tpm1* and *Tpm2* are involved in regulating EMT and stabilizing actin stress fibers [68]. Both *Tpm1* and *Tpm2* are induced by TGFβ-2 during EMT in LECs and have been used as EMT markers in lens cells [68,132]. Thus, the overexpression of decorin in the lens may suppress TGFβ-mediated EMT. Moreover, FGF2 also acts as an antagonist of TGF-β-mediated EMT [68]. We reported that *Tpm1* and *Tpm2* expression was repressed after adding FGF2 to cultured MLECs and HLECs [133]. In a previous in vivo functional study, the extent of decorin binding to TGF-β inhibits hepatic fibrosis [20]. In other words, there is a complex relationship between decorin and TGF-β in that TGF-β can mutually repress decorin gene expression in fibroblasts and LECs. This suggests that a complex feedback loop exists between these two factors to maintain cellular homeostasis after surgery and during wound healing (Figure 3). 

In a human PCO model using capsular bags from donor eyes that received an IOL, the levels of MMP2 and MMP9 were transiently elevated [134]. Additionally, pro-MMP2 and pro-MMP9 levels were increased and sustained upon treatment with active TGF-β2 [134]. Furthermore, the levels of pro-form MMP2 and MMP9 and active MMP2 were in-creased in the bathing media upon treatment with TGF-β in an organ-cultured lens from a rat ASF model [135]. In addition, the broad-spectrum MMP inhibitor GM6001 and a spe-cific MMP2/9 inhibitor suppressed the TGF-β induced-ASF [135]. From these studies, we speculate that decorin may be cleaved by MMPs in ASF and PCO and accelerate the acti-vation and release of TGF-β.

In the context of wound healing [6], it has been suggested that decorin interacts by localizing TGF-β, FGF2, and EGF close to the receptor. In the pathogenesis of PCO and other fibrotic eye diseases, decorin expression may be upregulated by FGF2, suppressing EMT and inducing normal cell proliferation and differentiation in LECs (Figure 3). We previously established a mouse PCO model with injury-induced EMT and fibrosis in the lens capsule to analyze the effect of decorin overexpression in wound healing in the lens using lens-specific human decorin conditional transgenic mice (*hDCN*-Tg) [7,133]. To examine the effect of decorin over-expression on the eye lens in vivo, we generated lens-specific *hDCN*-Tg using a plasmid in which Pax6-human αA-crystallin composite promoter (CPV14) [136] drives *hDCN* cDNA, as described in our previous study [7]. When the anterior capsule was injured via needle puncture, LECs differentiated into fibroblasts during the healing process via epithelial-myoblastic transition (EMyoT) [94,95] and transformed into αSMA-positive myofibroblasts. Wound healing was enhanced and fibrosis was suppressed after lens injury in *hDCN*-Tg mice compared with those in the wild-type (WT) [7]. It has been reported that cutaneous excisional wound healing in decorin-deficient mice (*Dcn*^−/−^) was delayed compared with that in WT (*Dcn*^+/+^) controls [6]. Thus, it is clear that the overexpression and addition of decorin may serve as an excellent therapeutic approach for improving wound healing and PCO. Moreover, the lenses of *hDCN*-Tg mice were clear and did not show any phenotypic abnormalities, such as abnormal arrangement or swelling of lens fibers and cell loss or abnormal growth of LECs through 48 weeks of age (Figure 4). Hence, decorin does not adversely affect the normal development and differentiation of the mouse lens.

In another study on the lens, decorin was found to inhibit high glucose-induced ROS production and apoptosis by reducing the induction of p22phox and phospho-p38 in a cultured human LEC cell line [137]. It was also reported that decorin could increase the levels of endogenous antioxidants, such as glutathione peroxidase and superoxide dis-mutase (SOD), and decrease the levels of tissue malondialdehyde, nitric oxide, and caspase-3 in the rat cerebrum after post-traumatic brain injuries [138]. Furthermore, decorin could protect retinal pigment epithelial (RPE) cells from H2O2-induced oxidative stress and apoptosis by promoting autophagy [11]. Furthermore, ROS induces the secretion and activation of TGF-β1 and is essential for TGF-β-mediated myofibroblast differentiation under different pathological conditions in various cells [56,139,140,141]. TGF-β also increases intracellular ROS content in different cell types, including LECs and trabecular meshwork (TM) cells by downregulating the expression of antioxidant-related genes, including *SOD*, *catalase*, *glutaredoxin*, and *lens epithelium-derived growth factor (LEDGF)* [139,142,143,144]. This redox imbalance is an important contributor to the pathophysiological effects of TGF-β, including fibrosis [145]. Thus, decorin may work as an antioxidant and anti-TGFβ1 agent and maintain redox balance in lens tissues. 

## 6. The Role of Decorin in Other Eye Diseases

### 6.1. Cornea

There are many reports on the role and function of decorin in the cornea. Decorin was first identified in the chicken cornea [146]. In humans, mutations in the decorin gene could cause congenital stromal corneal dystrophy (CSCD) when truncated decorin binds to collagen, resulting in aggregation in the stroma [100,147]. CSCD is the only human disease associated with mutations in decorin. CSCD is a rare autosomal dominant disorder characterized by small opacities found throughout the stroma [100,148]. Decorin is believed to inhibit corneal neovascularization under normal physiological conditions and is expected to maintain avascularity in the cornea [29]. The overexpression of decorin in cultured human corneal fibroblasts inhibited the TGF-β-induced expression of fibrogenic ECM genes such as fibronectin; alpha collagen type-I, -III, and -IV; and αSMA, a known EMT marker. In addition, the overexpression of decorin via tissue-targeted adeno-associated virus (AAV) 5-mediated delivery of the Decorin (*Dcn*) gene reduced corneal fibrosis caused by photorefractive keratectomy and corneal neovascularization in rabbits in vivo [149,150]. In addition, AAV5–*Dcn* gene delivery in the rabbit cornea revealed that it is safe and non-toxic to the eye in vivo during the 6-month follow-up period [151]. Thus, decorin plays a significant role in maintaining corneal transparency. 

Decorin has also been reported to have therapeutic applications for various diseases of the cornea. For example, sustained topical delivery of a fluid gel eye drop containing human decorin to the cornea promoted wound healing and reduced corneal scarring and opacity in a murine model of pseudomonas-induced keratitis [152]. Furthermore, topical decorin treatment also promoted corneal epithelial nerve regeneration after corneal abra-sion injury using Cx3cr1-deficient mice spontaneously lacking resident corneal epithelial dendritic cells [153]. From these results, it was hypothesized that decorin induces corneal nerve regeneration through the activation of dendritic cells [153]. Thus, decorin gene therapy or the optical administration of decorin has a high potential for treating corneal fibrotic and angiogenic diseases.

### 6.2. Glaucoma

Intraocular pressure (IOP) is the cardinal modifiable important risk factor for the on-set and progression of glaucoma [154,155]. TM and Schlemm’s canal provide the major route for the outflow of the aqueous humor from the anterior segment of the eye and are responsible for the raised IOP associated with POAG due to increased outflow resistance [156]. Here, the TM exhibits marked contractile properties, and TM contraction is associ-ated with reduced aqueous humor drainage through the outflow system [157]. Decorin has extenisve anti-fibrotic effects in glaucoma. The aqueous level of decorin was found to be decreased in eyes with glaucoma compared with non-glaucomatous eyes [88]. Moreo-ver, it has been reported that soluble decorin attenuated the generation and expression of typical downstream target genes, such as *CCN2/CTGF*, *Fn*, and *Col IV*, in human TM cells by TGF-β2. The negative reciprocal regulation of decorin and TGF-β was also observed, confirming the dramatic downregulation of decorin in the canonical outflow pathway of patients with POAG [14]. Expression of decorin mRNA was also decreased in the endo-thelial cells of the Schlemm canal derived from glaucomatous patients [158]. Intracameral injection of human recombinant decorin considerably reduced fibrosis in the TM and IOP and inhibited retinal ganglion cell loss in a rodent model of TM fibrosis via repeated in-tracameral injection of TGF-β [8]. Thus, decorin may be useful in developing an effective therapy for eyes with ocular hypertension and POAG associated with the fibrosis of the TM.

In POAG, the disturbed ECM metabolism and ECM remodeling are observed in the human optic nerve (ON) head, which is particularly obvious in the lamina cribrosa (LC) layer [159]. The remodeling processes in the ON head of glaucomatous patients are ac-companied by a disruption of the homeostatic balance of growth factors such as TGF-β [160]. In oligonucleotide microarray analysis using cultured LC cells from POAG human donors, expression of decorin was decreased [159]. In decorin-deficient mice, it was re-ported that an increase in IOP and a loss of ON axons [14] as well as a dramatic upreg-ulation of CTGF/CCN2, TGF-β-1, and TGF-β-2 were found [161]. In ON astrocytes, decorin negativery regulated TGF-β and CTGF/CCN2 [161]. Thus, the downregulation of decorin in the glaucomatous Schlemm canal endothelial cells, LC cells, and optic nerve head may affect the progression of POAG. The treatment of decorin may attenuate the remodeling process in the ON head observed during the progression of glaucoma and may reduce IOP.

Furthermore, in patients with uncontrolled glaucoma with progressive visual field loss due to glaucoma medication, IOP reduction by glaucoma surgery may be necessary. Glaucoma filtration surgery is a useful technique in reducing the IOP. Although this tech-nique immediately reduces IOP, the long-term effect is often eclipsed by the postoperative wound-healing process [162,163,164]. After glaucoma filtration treatments, the subconjuncti-val injection of decorin in rabbits showed markedly reduced conjunctival scarring and maintained lower IOP levels [165]. Thus, the application of decorin can inhibit the ob-struction of the fistula, which in turn inhibits fibroproliferative processes in fibroblasts from Tenon’s capsule and the episclera after filtration surgery [165].

The protein expression of decorin was also found to be increased in LECs of patients with PXG compared with those with pseudoexfoliation syndrome [106]. Simultaneously, TGFβ1 and FGF2 expression is elevated in the LECs of patients with PXG [106,166]. Thus, through the protein interaction networks of TGF-β and FGF2 that induce ECM remodeling, decorin may play a potential role in PXG and may be used as a novel therapeutic target for glaucoma.

### 6.3. Retinal Diseases 

SLRPs are crucial in many important biological functions involved in the mainte-nance of retinal homeostasis [13]. Decorin localizes in all layers of the human retina, in-cluding the RPE, Bruch’s membrane, choroidal stroma, and choroidal blood vessel walls [12]. A key step in PDR is neo-vascularization, beginning with the activation of the angio-genic switch after ischemic changes. Wet AMD is characterized by the so-called angio-genesis choroidal neovascularization or retinal aingiomatous proliferation [167]. Decorin is a key regulator for angiogenic responses by acting as an inhibitor for multiple RTKs and suppressing VEGFR through various mechanisms in choroid-retinal endothelial cells [26,168]. Moreover, many reports have demonstrated the anti-angiogenic effects of decorin in the retina [168,169]. A decreased expression of decorin has been reported in the retina of animal models of ischemic disease in the retinal vessels, such as oxygen-induced retinopathy in rats [170,171]. In another study, decorin treatment in rats with oxygen-induced retinopathy inhibited retinal neovascularization by reducing neovascular endothelial cell proliferation and the immunoreactivity of VEGF and TNF-α in the retina [172]. Additionally, intravitreal injection of decorin extensively suppressed VEGF and TNF-α expression and inhibited TGF-β expression in the RPE-choroid complex in a mouse model of laser-induced choroidal neovascularization, a model of AMD [169]. Thus, the administration of decorin into the eyes may inhibit VEGF-induced retinal diseases related to angiogenesis. Moreover, PVR is a complex process involving cell proliferation and ECM remodeling [173]. TGF-β is over-expressed in the vitreous humor of patients with PVR and PDR [55]. TGF-β is also presumed to be involved in the contraction of proliferative membranes and subretinal strands in PVR [174]. In an experimental model of rabbit traumatic PVR, intravitreal injection of decorin during vitrectomy reduced the development of fibrosis and tractional retinal detachment [175]. Therefore, decorin is also expected to prevent or regress neovascularization, EMT, and fibrosis in ischemic retinal diseases such as retinopathy of prematurity, proliferative DR, PVR, retinal detachment, and AMD as a TGF-β antagonist [9,11,107,168,172,175]. 

## 7. Summary

In summary, decorin can neutralize several important mediators of fibrosis and an-giogenesis in the eyes. Decorin has been shown to have anti-fibrotic effects in various an-imal models of eye diseases, and no toxicity has been reported in these studies. Decorin regulates the activity of multiple growth factors, receptor tyrosine kinases, and extracellu-lar matrix components. Although preclinical studies have been conducted, decorin has not yet been clinically applied as a drug due to its short half-life in circulation and diffi-culty in mass production since high doses are required for effective treatment. In the future, it is expected that GAG-unbound decorin core proteins, which are easier to create, will be produced in large quantities for use in pharmaceuticals [27,176]. Therefore, decorin is expected to be clinically used in the treatment of various ocular diseases needing adjuvant peptide therapies, including PCO, due to its anti-fibrotic, anti-inflammatory, anti-angiogenic, and antioxidant effects by affecting several growth factor pathways in the eyes. Hence, the development of effective intraocular delivery methods for decorin is warranted.

## Figures and Tables

**Figure 1 cells-12-00074-f001:**
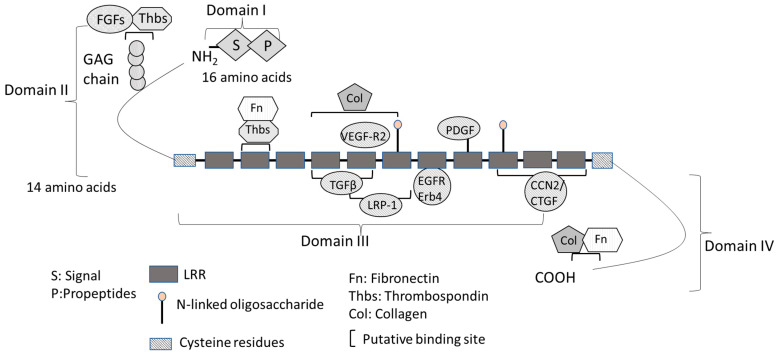
Structural and putative functional features of the decorin (DCN) molecule. Mammalian decorin contains core proteins and a glycosaminoglycan (GAG) side chain attached to a serine residue near the N terminus. Four domains of core proteins (I–IV) of DCN are shown. In domain I, the initial 16-amino-acid long signal peptide directs the incipient core protein to the rough endoplasmic reticulum and is cleaved via the co-translational pathway. In domain II, the propeptide is composed of 14 amino acids and regulates the attachment of the GAG chain. The protein core consists of tandem leucine-rich repeats (LRRs) flanking cysteine-rich disulfide domains. Twelve LRRs in domain III comprise the major central domain. Most decorin ligands share an interface with decorin at its core protein, an important hub for the extracellular matrix, cell surface receptors, and growth factors. Decorin has a high-affinity binding site for collagen (Col) at LRRs 4–6 and a low-affinity site at the C-terminus. The binding sites of TGF-β isoforms are LRR4 and LRR5 at the decorin core protein in domain II. The LRR7 domain binds to EGFR and ErB4 and activates the MAPK pathway. Decorin interacts with multiple growth factor signaling pathways. The carboxyl terminus domain and binding sites for fibronectin Fn and collagen Col can be found in domain IV.

**Figure 2 cells-12-00074-f002:**
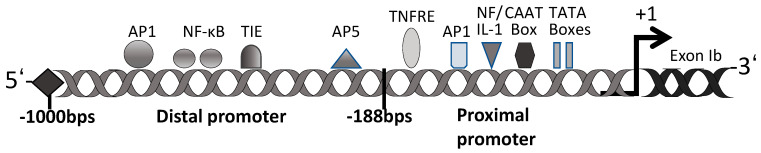
Schematic representation of the human decorin promoter. Consensus sequences for cis-regulatory elements are labeled.

**Figure 3 cells-12-00074-f003:**
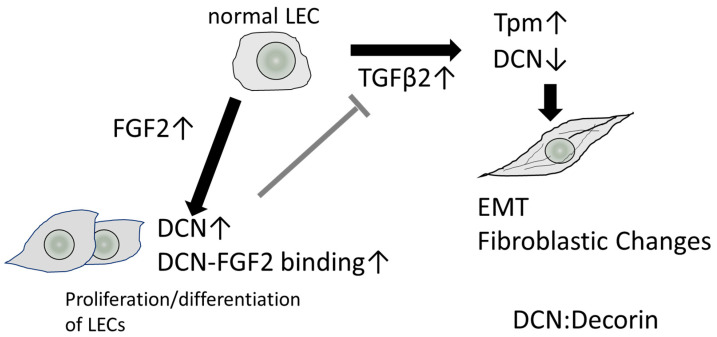
Schematic representation of the role of decorin (DCN) in the LECs. FGF2 induces the expression of DCN, but its mechanism is unclear. FGF2 binds to the GAG chain of DCN enhancing the activity of FGF2 and potentially promoting LEC proliferation and differentiation. Higher levels of DCN bind to TGF-β and inhibit its activity. On the other hand, TGF-β binds to the promoter of DCN and represses decorin expression, possibly promoting EMT and fibrosis. Increased expression of decorin inhibits the expression of Tpm1 and Tpm2 and may promote normal cell growth and wound healing. DCN, decorin; LECs, lens epithelial cells; FGF2, fibroblast growth factor 2; GAG, glycosamino-glycan; TGF-β, transforming growth factor-β; EMT, epithelial–mesenchymal transition.

**Figure 4 cells-12-00074-f004:**
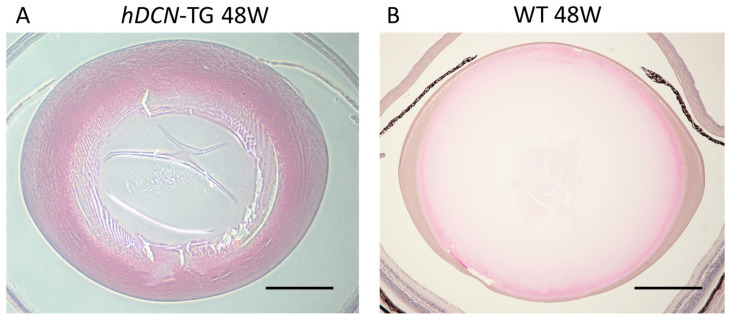
Histochemical analysis of lenses in 48-week-old (W) *hDCN*-Tg mice. Adult lenses in 48 W *hDCN*-Tg animals (**A**) did not show lens fiber damage and pathological changes and did not differ from those in WT mice (**B**). Overexpression of *hDCN* did not affect the histological changes in lens devel-opment. Scale bar = 200 µm.

## Data Availability

Not applicable.

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
