# Peer review of "Role of Decorin in the Lens and Ocular Diseases"

_cells, 2022, doi:10.3390/cells12010074_

Round 1

Reviewer 1 Report

Kubo et al. have submitted a very interesting review about a relevant topic with the title “Role of decorin in the lens and eye diseases”. The manuscript elucidates the function of decorin in the different ocular tissues and links the presence or absence of decorin to different ocular diseases. The authors nicely describe the interaction of decorin with different growth factors and their pathways and they give a substantial outlook about the potential therapeutic benefits. There are some minor issues that should be addressed prior of publication:

1.     Title: The title should be altered as the lens is a part of the eye. They could hint that they will mainly focus on the lens

2.     Introduction (line 44): The CCN society already declared years ago, that the designation CTGF is misleading, so the nomenclature was altered to CCN2. It is allowed to write CCN2/CTGF, but CTGF alone must be avoided.

3.     Structure and function of decorin (line 98): CCN2/CTGF in not only a TGF beta inducible factor. It is an matricellular protein, which can be regulated by many different factors.

4.     Interaction between decorin and TGF beta (line 113): In the ocular tissues, TGF beta 2 has many different functions, so that it is too simple to describe it only as a profibrotic factor. 

5.     Interaction between decorin and TGF beta (line 119): The Smad pathway has more functions than the fibrotic disease.

6.     Interaction between decorin and TGF beta (line 126): Smads have to be changed into Smad2/3 as there are many Smad proteins.

7.     Interaction between decorin and TGF beta (line 131):  “For the activation of TGF beta” is misleading as the meaning of the sentence is not about the activation, it is about the TGF beta effect on different pathways.  

8.     Interaction between decorin and TGF beta (line 141-162):  A schematic drawing of the promoter region and the different transcription modulators would be helpful.

9.     Interaction between decorin and TGF beta (line 159):  Change “andfter” to “and after”

10. Interaction between decorin and TGF beta (line 162-181):  TGF beta has also positive effects in the eye like maintaining the immune privilege. The authors should also address the positive role of TGF beta.

11. Interaction between decorin and TGF beta (line 177):  The aqueous humor analysis was performed not in comparison to healthy subjects, it was in comparison to cataract patients.

12. Distribution of decorin in the eye:  The authors should describe the presence of the decorin in all ocular tissues in this chapter like retina and optic nerve. They describe the presence of decorin in retina in a following chapter, but it should be added in the distribution part.

13. The role of decorin in the lens (line 249 to 252):  A brief description about the function of TPM 1 and 2 should be added.

14. The role of decorin in the lens (line 278):  Change “EGFclose” to EGF close

15. The role of decorin in the lens (line 283): A brief description of the conditional model should be added.

16. Glaucoma (line 348): The major risk factor is not a pathologic elevated IOP. All the clinical studies revealed that IOP in general is a risk factor. 

17. Glaucoma (line 360):  Reduced levels of decorin in the endothelial cells of the Schlemm canal derived from glaucomatous patients were also observed (Overby et al. 2014).

18. Glaucoma: The function of decorin in the optic nerve head and in lamina cribrosa cells is missing.

Reviewer 2 Report

The review article “Role of decorin in the lens and eye diseases” by Kubo et al., (2025539) discusses the role of the small leucine-rich proteoglycan decorin with respect to wound healing and EMT/PCO in the lens and eye diseases.

In this review, the authors describe general features of decorin and the regulation of decorin expression.  They also review the literature with respect to how decorin may regulateseveral growth factors, but particularly TGF-β and FGF signaling. I have several suggestion and comments that I believe would improve the manuscript. I have listed these in general order in which they appear in the text and not by the order of importance.

1.       Line 41: Add “with” in “Decorin interacts with and modulates multiple growth….”

2.       Line 44: What does it mean to directly block various receptor tyrosine kinase (RTK) family proteins? Do the authors mean to say that decorin inhibits the activation of RTKs? The wording is confusing here.  Also, Toll-like receptors are not receptor tyrosine kinases.

3.       In line 64 and in figure 1, the authors mention that FGF-2 binds to the GAG chain of decorin.  However, in line 66, the authors state that FGF1, 2, 7 and 8 bind to the GAG chain of decorin. Wouldn’t it be better to just state in line 64 and in the figure that the GAG side chain binds to FGFs?

4.       Line 73: I do not believe that incioient is a word.  What do the authors mean here?

5.       Line 74 “and is cleaved in cotranslation..”  I believe that the authors mean “and is cleaved via the co-translational pathway.”

6.       Line 86: Add “is”  ….rich domains and is important for its….

7.       Line 88: I believe that the authors mean to say “at the C-terminus for interacting with” or “at the C-terminus that interacts with”

8.       Lines 113-114: Replace “Expressions of TGF-β are increased” with “The expression of TGF-β is increased”.

9.       LIne 118: Doesn’t TGF-β increase the production of some ECM components, including fibronectin and some collagens? The text states that TGF-β inhibits the production of ECM.

10.   Lines 129-131: The sentence that starts “Besides Smad pathways, other..are also crucial for the activation of TGF-β.”  This sentence could be improved by not referring to the “activation of TGF-β”, but rather focus on the action or the effects of TGF-β.  TGF-β is produced in a latent form that needs to be activated for receptor binding….I don’t believe that this is what the authors are referring to in this sentence.

11.   The review tends to be redundant/repetitive in several instances. For example, the authors mention the low affinity binding site for collagen in the C-terminal domain in lines 87-89 and then again in lines 102 and 103; the first sentence in section 3 (line 113) is “TGF-β is considered a potent and ubiquitous pro-fibrotic cytokine” and this is followed in the same section in line 133-134 “Thus fibrosis is also regulated and induced by TGF-β signaling.”; on line 137-138 the authors state “In addition, decorin can bind to TGF-β and inhibit the binding of TGF-β ligands to their receptors.” whereas in lines 139-140 the authors continue “The bound complex formation with deorin prevents the binding of TGF-β to its receptor, repressing its activity.”  While a little repetition is OK, there seems to be a lot of it in this review. The authors should edit the review to prevent unnecessary redundancy.

12.   Lines 144-145: “The promoter of decorin is divided into two main regions: a proximal promoter at 188 bp and a distal promoter at 800 bp.” Do the authors mean that the proximal promoter is 188 bp upstream of the transcription start site (-188 to +1) and the distal promoter is 800 bp upstream of the transcription start site (-800 to -188)?  The authors should be more precise to avoid confusion. This description of the decorin control regions could benefit from a diagram.

13.   Lines 184-186 “In the eyes, decorin is expressed in the corneal stroma, conjunctival stroma, trabecular meshwork, lens, and in all layers of the retina and sclera, and they are secreted in the tears, aqueous humor, and vitreous humor”  Does the they refer to decorin? If so, they are should be replaced by it is.

14.   Lines 189=190: Decorin expression cannot be upregulated in the lens capsule.  The lens capsule is an acellular extracellular matrix.  While decorin levels can be increased under certain conditions. The upregulation has to be due to the cell that secrete decorin.

15.   Line 194, replace “preeviously” with previously.

16.   Lines 212-214: “After cataract surgery, fibroblasts are activated in residual LECs and the lens 212 capsule, expressing high levels of fibronectin, procollagen, Tpm, and αSMA, resembling 213 the wound healing process.”  This sentence makes no sense.  Fibroblasts are activated in residual LECs?  Does this mean that lens epithelial cell harbor latent, internal fibroblasts????  And again, the lens capsule doesn’t actively express anything.

17.   Do the authors suggest that when TGF-β binds to decorin that it’s activity is reduced but when FGF2 binds to decorin, its activity is potentiated? The authors might want to explain this dichotomy.

18.   Line 374: “The protein expression of decorin was also found to be increased in the lens capsule”  While the levels of decorin protein may be higher in the lens capsule of PXG patients, the lens capsule cannot express decorin.

19.   Line 410:  The authors mention that decorin has been shown to have antitumor effects in various animal models of eye diseases.  I am unaware of any specific mention in this review of an animal model of eye disease where decorin has shown to be inhibitory to ocular tumors.  All of the mentions in the text are of general antitumor properties of decorin. If the authors want to make this statement, they should cite specific examples.   

Line 413: What specifically do the authors mean when they say that decorin “serves as a guardian from matrix?” The authors need to be very clear here.  
